# Transcriptomic Response of Differentiating Porcine Myotubes to Thermal Stress and Donor Piglet Age

**DOI:** 10.3390/ijms241713599

**Published:** 2023-09-02

**Authors:** Fabio Sarais, Katharina Metzger, Frieder Hadlich, Claudia Kalbe, Siriluck Ponsuksili

**Affiliations:** 1Research Institute for Farm Animal Biology (FBN), Institute of Genome Biology, 18196 Dummerstorf, Germany; sarais@fbn-dummerstorf.de (F.S.); hadlich@fbn-dummerstorf.de (F.H.); 2Research Institute for Farm Animal Biology (FBN), Institute of Muscle Biology and Growth, 18196 Dummerstorf, Germany; metzger@fbn-dummerstorf.de (K.M.); kalbe@fbn-dummerstorf.de (C.K.)

**Keywords:** myotubes, differentiation, pig, transcriptome, thermal stress, skeletal muscles

## Abstract

Climate change is a current concern that directly and indirectly affects agriculture, especially the livestock sector. Neonatal piglets have a limited thermoregulatory capacity and are particularly stressed by ambient temperatures outside their optimal physiological range, which has a major impact on their survival rate. In this study, we focused on the effects of thermal stress (35 °C, 39 °C, and 41 °C compared to 37 °C) on differentiating myotubes derived from the satellite cells of *Musculus rhomboideus*, isolated from two different developmental stages of thermolabile 5-day-old (p5) and thermostable 20-day-old piglets (p20). Analysis revealed statistically significant differential expression genes (DEGs) between the different cultivation temperatures, with a higher number of genes responding to cold treatment. These DEGs were involved in the macromolecule degradation and actin kinase cytoskeleton categories and were observed at lower temperatures (35 °C), whereas at higher temperatures (39 °C and 41 °C), the protein transport system, endoplasmic reticulum system, and ATP activity were more pronounced. Gene expression profiling of *HSP* and *RBM* gene families, which are commonly associated with cold and heat responses, exhibited a pattern dependent on temperature variability. Moreover, thermal stress exhibited an inhibitory effect on cell cycle, with a more pronounced downregulation during cold stress driven by *ADGR* genes. Additionally, our analysis revealed DEGs from donors with an undeveloped thermoregulation capacity (p5) and those with a fully developed thermoregulation capacity (p20) under various cultivation temperature. The highest number of DEGs and significant GO terms was observed under temperatures of 35 °C and 37 °C. In particular, under 35 °C, the DEGs were enriched in insulin, thyroid hormone, and calcium signaling pathways. This result suggests that the different thermoregulatory capacities of the donor piglets determined the ability of the primary muscle cell culture to differentiate into myotubes at different temperatures. This work sheds new light on the underlying molecular mechanisms that govern piglet differentiating myotube response to thermal stress and can be leveraged to develop effective thermal management strategies to enhance skeletal muscle growth.

## 1. Introduction

In recent years, concern over global warming and climate change has increased, and one of the main areas of agriculture that will be impacted is the livestock sector [1]. The swine industry incurs a significant cost from heat stress (HS), which can be caused by an absence of functioning sweat glands and a thick layer of subcutaneous adipose tissue that produces a significant amount of metabolic heat [2]. Heat stress may lead to significant alterations in a swine’s physiology and metabolism, which have a detrimental impact on a number of production factors, such as growth, feed intake and conversion efficiency, carcass features, immunological function, and meat output and quality [3].

As a counterpart, cold stress (CS) lowers young animals’ antioxidant capacity and immune system in addition to affecting their growth rate [4,5]. Mammals in cold climates primarily use two methods for producing heat to keep their body temperatures stable: shivering thermogenesis (ST) and non-shivering thermogenesis (NST). ST is the process of producing heat via the contraction of skeletal muscles, whereas non-shivering thermogenesis is the process of generating heat through food metabolism [6,7]. Therefore, due to a lack of brown adipose tissue, newborn piglets rely almost entirely on ST [8]. Thus, heat production mainly depends on the oxidation of specific substrates such as carbohydrates, lipids, and proteins, which are supplied to skeletal muscles with a precise timing and quantity [9]. The mature muscle cells that serve as the primary structural component of skeletal muscle play a central role in the thermoregulatory system of newborn piglets [10]. Cold exposure affects muscle maturation, increases the expression of type I *MyHC* and decreases that of fetal *MyHC*, and increases the activities of oxidative enzymes [10]. Multinucleated myotubes are formed through differentiation of myoblasts following withdrawal from the cell cycle and expression of myogenic regulatory factors (MRFs) [11]. MRFs consist of four muscle-specific proteins: myoblast determination factor (MyoD), myogenic factor 5 (Myf5), myogenin (MyoG), and myogenic regulatory factor 4 (Mrf4) [12]. These proteins cooperate to establish the muscle phenotype by regulating proliferation, irreversible cell cycle arrest, and regulated transcription of sarcomeric and muscle-specific genes for differentiation and sarcomere formation [13]. Our previous research highlighted the differential expression of numerous genes in porcine myoblast cultures subjected to heat or cold stress, with these transcripts linked to a range of biological processes and signaling pathways [14]. Notably, the majority of molecular changes occurred at the control temperature of 37 °C, which is the standard cultivation temperature for porcine muscle cell cultures and appropriate for comparing myoblast cells from piglet donors that are still developing their thermoregulatory capabilities under temperature stress [14]. We applied our previously established cell pooling method to account for the potential impact of both temperature and donor cell age on myotube induction [15]. 

In this study, we used myoblasts from piglets aged 5 and 20 days, cultured, and induced myotubes formation across multiple temperatures (35, 37, 39, and 41 °C), with a particular interest in examining changes in molecular processes, energy metabolism, and oxidative stress. Through this research, we aimed to evaluate whether the developmental commitment of muscle cells, particularly during myotubes differentiation, influences their responsiveness to thermal stress and alters the related biological pathways.

## 2. Results

### 2.1. Transcriptome Analysis in Response to Culture Temperatures

The transcriptome’s response to different culture temperatures, compared to the standard culture temperature for porcine muscle cells (37 °C), revealed 4754 differentially expressed genes (DEGs) at 35 °C, of which 2262 were upregulated and 2492 downregulated. At 39 °C, 2892 DEGs were recorded, of which 1560 were upregulated and 1332 downregulated. At 41 °C, 1885 genes were upregulated and 1582 downregulated, for a total of 3467 DEGs. A total of 1363 DEGs overlapped between the different culture temperatures (Figure 1a and Appendix A).

DEGs were further used for gene ontology (GO) and Kyoto Encyclopedia of Genes and Genomes (KEGG) pathway enrichment analyses (Figure 2a and Appendix A). Cellular response to peptide, supramolecular fiber organization, and intracellular transport were observed from enriched DEGs in a biological process (BP) for the three different experimental temperatures. A lower temperature than physiological (35 °C) demonstrated that the enriched DEGs were correlated to biological pathways as a cellular protein catabolic process, proteolysis involved in cellular protein catabolic process, regulation of cell cycles, and actin filament-based process. Temperatures above the standard culture temperature (39 °C and 41 °C) showed the enrichment of DEGs with biological processes linked to actin filament-based processes, intracellular protein transport, intracellular protein transport, and Golgi vesicle transport. The enriched DEGs for cellular compounds (CC) were associated with the cell leading edge, sarcolemma, ruffle, Golgi apparatus, transferase complex, focal adhesion, endoplasmic reticulum, cell-substrate junction, and bounding membrane of organelle at all three experimental temperatures. Sarcolemma was shown to be the unique CC linked to the enriched DEGs at 35 °C. The differential expressed genes in the 35 °C vs. 37 °C and 41 °C vs. 37 °C comparison showed a correlation to peptidase complex. Higher temperatures (39 °C and 41 °C) showed enriched DEGs associated with CC, such as the nuclear outer membrane-endoplasmic reticulum membrane network and endoplasmic reticulum. With regard to molecular functions (MF), we observed an enrichment in DEGs related to protein kinase activity, kinase activity, cytoskeletal protein binding, protein domain specific binding, and RNA binding at all the experimental temperatures. Comparison of 35 °C vs. 37 °C showed DEGs linked to phosphotransferase activity and alcohol group as acceptor. The enriched DEGs in comparisons between 35 °C and 37 °C and 39 °C vs. 37 °C shared MF associated with protein-containing complex binding and identical protein binding. Moreover, comparisons between 35 °C and 37 °C, and 41 °C and 37 °C, revealed enriched DEGs coupled with transferase activity, transferring phosphorus-containing groups, and acting binding. Additionally, the structural constituent of cytoskeleton was linked to the enriched DEGs reported in the 39 °C vs. 37 °C comparison. Overall, most changes in the macromolecule degradation and actin kinase cytoskeleton categories were observed at lower temperatures (35 °C), whereas at higher temperatures (39 °C and 41 °C), the protein transport system, endoplasmic reticulum system, and ATP activity were more pronounced (Figure 2a).

KEGG analysis of enriched DEGs in myotubes cultured at the different temperatures revealed a linkage with protein processing in the endoplasmic reticulum, cell cycle, focal adhesion, and regulation of actin cytoskeleton for the analyzed temperature clusters. The comparison of 35 °C vs. 37 °C revealed enriched DEGs related to the focal adhesion, PI3K-Akt signalling pathway, Axon guidance, HIF-1 signaling pathway, mTOR signaling pathway, p53, and Wnt signaling pathway (Figure 2b). KEGGs pathways such as Ferroptosis and cellular senescence were reported for enriched DEGs at 35 °C and 41 °C. Additionally, experimental temperatures of 35 °C and 39 °C showed enriched DEGs related to ubiquitin-mediated proteolysis and thyroid hormone signaling. Moreover, several specific genes reported to be involved in cold stress response, i.e., regulatory genes for cell cycle (*p53/CDNK*), heat shock proteins (*HSPs*), development regulator genes, signal transduction gene markers, and RNA binding protein family (*RBM*) showed generally different expression patterns, in a temperature-dependent manner (Figure 3 and Appendix A).

Experimental temperatures above the standard culture temperature (39 °C and 41 °C) shared DEGs linked to the KEGGs pathways, such as protein export, EGFR tyrosine kinase inhibitor resistance, and mRNA surveillance pathways. Additionally, enriched DEGs reported in the 39 °C vs. 37 °C comparison showed a linkage to KEGGs pathways such as hypertrophic cardiomyopathy, insulin signaling pathway, AMPK signaling pathway, dilated cardiomyopathy, and lysosome. Furthermore, particular enriched DEGs between 41 °C vs. 37 °C were associated with KEGG pathways including nucleocytoplasmic transport, spliceosome, carbon metabolism, and TCA cycle (Figure 2b).

### 2.2. Transcriptome Analysis in Response to the Age of Donor Piglets to Culture Temperatures

In total, 1101 DEGs were identified between the two groups (p5 and p20), of which 527 were upregulated and 574 downregulated (Appendix A). When comparing p5 with p20, a total of 2844 transcripts were found to be significant in at least one combination of each temperature (Appendix A). The transcriptome’s response to different culture temperatures of myotubes from donor piglets (p5 and p20) revealed 825 differential expressed genes (DEGs) at 35 °C, of which 366 were upregulated and 459 downregulated (Figure 4a). At 37 °C, 794 DEGs were recorded, of which 380 were upregulated and 414 downregulated (Figure 4b). At 39 °C, 612 genes were upregulated and 275 downregulated, for a total of 887 DEGs (Figure 4c). At 41 °C, 231 genes were upregulated and 443 downregulated, for a total of 674 DEGs (Figure 4d). A set of 72 genes, representing 3% of the DEGs, overlapped in the five comparisons (Figure 1b) as being independent of the change in temperature.

GO and KEGG pathway enrichment analyses were performed using DEGs from each experimental setup. GO analysis between the two donor groups revealed a wide variety of BPs, CCs, and MFs (Appendix A). A temperature-independent GO analysis of the enriched DEGs between the two donor groups, p5 and p20, reported biological pathways related to muscle cell differentiation, regulation of anatomical structure morphogenesis, tube morphogenesis, circulatory system development, tissue migration, striated muscle tissue development, myofibril assembly and positive regulation of transferase activity (Figure 5a). At 35 °C and 37 °C, the enriched DEGs for biological processes (BP) were associated with muscle cell differentiation, regulation of anatomical structure morphogenesis, anatomical structure formation involved in morphogenesis, circulatory system morphogenesis, tissue migration, striated muscle tissue development, myofibril assembly, and muscle differentiation. Tube morphogenesis and striated muscle cell differentiation were only reported for DEGs at 35 °C and 37 °C, respectively. Temperatures above the standard (39 °C) reported a set of DEGs linked to ribosome, myofibril assembly, positive regulation of transferase activity, peptide metabolic process, translation, striated muscle cell development, tube formation, and positive regulation of kinase activity. Comparing p5 and p20, few changes in biological processes were observed at 41 °C.

KEGG analysis of enriched DEGs in cultured myotubes from the two donor groups (p5 and p20) at the different experimental temperatures (Figure 5b) revealed linkages with hypertrophic cardiomyopathy and dilated cardiomyopathy for the analyzed temperature clusters. KEGG analysis of the enriched DEGs between the two donor groups showed several pathways shared at the different temperatures analyzed. Enriched DEGs at 35 °C, 37 °C, and 39 °C involved KEGG pathways such as arrhythmogenic right ventricular cardiomyopathy, focal adhesion, cardiac muscle contraction, and adrenergic signalling in cardiomyocytes. Enriched DEGs at 35 °C and 37 °C included KEGG pathways such as the PI3KT-Akt signaling pathway, EGFR tyrosine kinase inhibitor resistance, and HIF-1 signaling pathway (Figure 6d). ECM-receptor interaction, tight junction, regulation of actin cytoskeleton, FC gamma R-mediate phagocytosis, and FoxO signaling pathways were reported as KEGG pathways for enriched DEGs at 37 °C. Furthermore, enriched DEGs at 37 °C and 39 °C showed common KEGG pathways, such as the adherens junction. Ferroptosis (Figure 6a) and autophagy (Figure 6c) were linked for enriched DEGs at 39 °C. 

A temperature-independent GO analysis of the enriched DEGs between the two donor groups, p5 and p20, reported KEGG pathways (Figure 6b) related to hypertrophic cardiomyopathy, dilated cardiomyopathy, adherens junction, arrhythmogenic right ventricular cardiomyopathy, focal adhesion, cardiac muscle contraction, adrenergic signalling in cardiomyocytes, PI3KT-Akt signaling pathway, ECM-receptor interaction, regulation of actin cytoskeleton, HIF-1 signaling pathway, FoxO signaling pathway, cellular senescence, TGF-beta signaling pathway (Figure 6b), and Leukocyte transendothelial migration.

In order to validate the microarray data, we selected 12 genes and measured their expression using qPCR (Appendix A, Figure 7). The microarray and qPCR data showed a high correlation based on Pearson’s correlation coefficient (*r*). The selected genes are involved in molecular pathways controlling various processes such as glycolysis/gluconeogenesis (*GPI, r* = 0.86, *p* < 0.05; *PGK1, r* = 0.95, *p* < 0.05), HIF-1 signaling pathway (*EGLN1 r* = 0.97, *p* < 0.05; *EGLN3 r* = 0.98, *p* < 0.001; *HIF1A r* = 0.98, *p* < 0.001), ferroptosis (*HMOX1, r* = 0.96, *p* < 0.005; *ACSL4 r* = 0.99 *p* < 0.001; *SCL93A14, r* = 0.89, *p* < 0.05), NF-kappa B signaling pathway (*NFKB1 r* = 0.96, *p* < 0.001; *NFKBIA r* = 0.99, *p* < 0.001), Thermogenesis (*NDUFS1 r* = 0.92, *p* < 0.05), and A Myogenic regulatory factor (*MYF5 r* = 0.74, *p* < 0.05).

### 2.3. GSH/GSSG Ratio in Response to Experimental Temperature Conditions

Glutathione has a protective role against the deleterious effect of reactive oxygen species (ROS) [16]. The ratio between GSH/GSSH is a valuable marker for oxidative stress and its homeostasis plays an important role in maintaining cellular redox status [17]. Our results showed that the GSH/GSSG ratio was affected by temperature (41 °C vs. 37 °C), but no significant differences were observed in the pool or pool/temperature comparison (Appendix A, Figure 8).

### 2.4. Metabolic Flux Assay

Analyses of the metabolic flux at the different temperatures were conducted, normalizing the oxygen consumption rate (OCR) values for the total protein content, as already described [18]. Cold and heat stress showed a similar result pattern for the different observed parameters, generally reduced compared to the control temperature. Statistical analysis demonstrated significant differences (*p* < 0.05) in non-mitochondrial oxygen consumption in heat-stressed cells (39 °C and 41 °C). Additionally, the coupling efficiency exhibited significant differences at 41 °C and between p5 and p20 pools. Detailed results and *p-values* are reported in Appendix A.

## 3. Discussion

The impact of HS and CS on newborn piglets affects their metabolism, immune system, differentiation and growth performances [19,20]. Interestingly, about 43% of the body mass is represented by the skeletal muscles, and they are key players as thermogenic and metabolic organs [21,22]. Several studies have already shown that the in vitro models of skeletal muscle frequently employ primary myotubes, which represent the basic structural element of skeletal muscle [23]. In our previous study on the transcriptomic profile of myoblasts derived from piglets of different ages, differences in thermoregulatory abilities under different temperatures were reported [14]. We are also interested in whether the thermal stress of myoblasts plays a role in the differentiation process by further inducing these myoblasts into myotubes. Therefore, the effects of age and temperature on myotube differentiation were further investigated in this study. Our aim was to decipher the effects of cold and heat stress in the primary differentiated myotubes of pigs derived from different piglets with different thermoregulatory abilities (p5 and p20).

### 3.1. Transcriptome Regulation in Response to Cold Stress 

Exposure of mammalian cells to cold temperatures generally decreases membrane transport, diffusion, and enzymatic reactions [24]. Interestingly, several mechanisms of cold stress response share similarities to those reported for heat stress response in mammalian cells [25], including perturbation in proteins homeostasis [26], reduction in cell cycle [27], damages at cytoskeleton level, and changes in membrane permeability [28]. To date, studies on the transcriptome during cold stress response for porcine myotubes have been scarce and limited to other skeletal muscle cell types. Interestingly, most DEGs were found under cold stress and were enriched in many important biological pathways, especially in the actin kinase cytoskeleton or focal adhesion. This result suggests that cold stress plays an important role in myotube differentiation. Our transcriptome analysis on differentiating porcine myotubes revealed several enriched biological process pathways connected to catabolic processes, regulation of cell cycle, and cytoskeleton modification. Additionally, metabolic function enriched pathways reported a high number of GO terms connected to kinase activity and protein binding. Interestingly, mRNA expression levels of the *ADGR* gene family were reported at lower levels (FC < −1.2) compared to the control group. The *ADGR* (Adhesion G protein-coupled receptors) gene family translates for a class of G protein-coupled receptors, involved in the regulation of the Wnt signaling pathway [29]. This result underpins our previous investigation on the C2C12 myoblast, in which Wnt pathway activation was altered during cold stress [30]. Additionally, expression levels of cell cycle gene markers such as the *TP53* and *CDKN* gene families were mainly upregulated during cold stress exposure, influencing the cellular proliferation [31,32]. Furthermore, the mRNA expression of the *HSP70* gene family was generally upregulated, especially the *HSPA4L* gene (FC > 3), probably shielding the cold-shocked cells from the negative consequences of protein aggregates [33]. Notably, the expression of *RMB* gene family exhibits upregulation in response to cold stress. Previous studies have extensively investigated this gene family and revealed its crucial role in cold stress response. Interestingly, the presence of an IRES (internal ribosome entry sequence) motif within a gene’s structure suggests its involvement in RNA-protein interactions [25,34,35]. Our KEGG analysis of enriched DEGs confirmed these findings, as we observed significant representation of mTOR signaling, Wnt signaling, p53 signaling, and cell cycle pathways associated with cold stress response. Specifically, the activation of mTOR signaling was implicated in the regulation of cellular differentiation in mesenchymal and satellite muscle cells [36]. Remarkably, cold stress response induced regulation of the Ferroptosis pathway, probably due to lipid peroxidation driven by the modulation of *MICU1* gene and *ACSL4*, which has been reported in several studies [37,38,39]. In comparing these results to our previous findings on cold-stressed myoblasts, we identified that the Cell Cycle pathway and RNA binding process were common between both cell types. However, it is worth noting that the *RMB3* and *ACSL4* gene expressions followed a similar regulation [14].

### 3.2. Transcriptome Regulation in Response to Heat Stress

Mild heat stress (39 °C) exposure of differentiating porcine myotubes mainly showed DEGs associated with catabolic processes, intracellular transports, actin filament-based processes, and supramolecular fiber organization. These results stand together with the identified KEGG pathways, which revealed activation of actin cytoskeleton, cell cycle, and a high impact in various signaling pathways, such as mTOR, insulin, and AMPK. An overall observation of the altered GO and KEGG pathways raises the hypothesis that mild heat stress exposure of differentiating myotubes induces cell morphology alteration, as reported in other cell types [30,40]. Interestingly, for enriched DEGs at 39 °C in the KEGG analysis, activation of the mRNA surveillance pathway was reported. This pathway is crucial for the rapid degradation of aberrant mRNA, which could be translated into shorter protein sequences due to a premature stop codon or in non-functional proteins due to incorrect mRNA sequences [41]. Notably, two members of the *HSP* gene family, *HSPB3* and *HSPBAP1,* showed an increased level of mRNA compare to the control group. These two genes have previously been reported to regulate myogenesis and cell growth, respectively [42,43]. Furthermore, these observations agree with our previous study on heat-stressed myoblasts [14]. 

Acute heat stress exposure (41 °C) of differentiating porcine myotubes revealed broadly the same GO terms as the mild heat stress group. Nevertheless, the mRNA expression of the *ADGR* gene family was higher than what we observed in the mild heat stress group. This might be correlated to enhanced signal transduction [44] and Wnt pathway activation [30]. Interestingly, the expression of regulatory genes for cell cycle (*p53/CDNK*) was lower than the other experimental temperatures, probably inhibited by the higher expression of *MDM2* [45]. Juxtaposing our findings with a former study on heat stressed myoblasts, we found that only a few GO terms, such as ribonucleoprotein complex and Golgi apparatus, were shared. Additionally, both analyses reported the involvement of the Cell Cycle KEGG pathway [14].

### 3.3. Transcriptome Regulation in the Different Developmental Stages of Donor Piglets at Different Experimental Temperatures

Undeniably, the first days after birth are critical for newly born piglets [46]. Although they endure temperatures that are much below their thermoneutral zone very soon after birth, newborn pigs’ adaptation to extrauterine life is a significant barrier to their survival and postnatal growth [47]. Additionally, the lack of brown adipose tissue, insufficient thermoregulation, reduced isolation, and rapid heat dissipation increase the risk of thermal stress [48,49,50]. Our study aimed to decipher the effects of different culture temperatures on the transcriptomic profiles of differentiating myotubes originating from piglets at different developmental stages. The experimental temperatures of 35 °C and 37 °C had the highest number of significant GO terms and showed some particularly specific pathways, such as insulin signaling, thyroid hormone signaling, and calcium signaling, which were only reported at 35 °C. The specific involvement of the above mentioned pathways could be explained by a precise response against cold stress, which appeared to increase insulin production, promoting sympathetically-mediated thermogenesis [51]; to increase thyroid hormones, in order to regulate metabolism, oxygen expenditure, and body temperature [52,53]; and to regulate Ca^2+^ levels, inducing hypothalamic thermoregulation [54]. Moreover, it is worth mentioning that there was a significant downregulation of the *ADIPOR1* and *ADIPOR2* genes between p5 and p20 at a temperature of 35 °C. These genes encode for receptors that are well-known for their interaction with adiponectin, a hormone family that plays a vital role in enhancing insulin sensitivity and regulating normal body temperatures during CS. These results suggest that the donor age of piglets, with a differential ability to thermoregulate under cold stress, plays an important role in myotube differentiation and temperature adaptation. However, at temperatures higher than 37 °C, only marginal variations between p5 and p20 were seen, mostly related to ribosomal and endoplasmic reticulum activity processes.

Interestingly, our analysis revealed the enrichment of differentially expressed genes (DEGs) in the ferroptosis pathways between p5 and p20 at temperatures of 35 °C and 39 °C. This finding aligns with a previous study conducted in myoblasts, which demonstrated that the regulation of the ferroptosis process occurs specifically at temperatures lower than the standard conditions [14]. These results suggest that the differentiation of myotubes, which is a stress-inducing process, is more pronounced at p5 compared to p20 even at mild heat temperatures, which is not the case for the proliferation of myoblasts. Importantly, the enrichment of autophagy at 39 °C when comparing p5 and p20 indicates the harmful consequences of elevated temperatures [55]. This finding further emphasizes the detrimental effects of temperature increase on cellular processes. Moreover, our analysis revealed that the HIF-1 signaling pathway and the TFG-β signaling pathway were specifically regulated during CS and at 37 °C. These pathways play essential roles in cellular adaptation and response to stress conditions, highlighting their importance for maintaining cellular homeostasis under challenging circumstances.

### 3.4. Effects of Thermal Stress on Metabolic Flux and GSH/GSSG Ratio

Metabolic flux, which is dependent on enzyme activity, substrate availability, and energy supply, is impacted by thermal stress [18]. While low temperatures slow down metabolic rates, high temperatures reduce enzyme activity and flux [56]. Thermal stress also changes the availability of substrates and the energy source, which impacts metabolic flow [57]. Our results showed a marked significant difference in the ratio of reduced glutathione (GSH) and oxidized glutathione (GSSG) at 41 °C compared to our controls (37 °C) (*p* < 0.001), due to a reduction in the GSH concentration. Multiple studies have shown that heat stress leads to a significant reduction in the levels of GSH [58,59,60]. The decrease in GSH is accompanied by an elevated production of reactive oxygen species (ROS), heightened susceptibility to high temperatures [61], and may promote cellular damage and oxidative stress, leading to regulation of the apoptotic process [62,63,64,65], which was highlighted in our enriched DEGs analysis. Furthermore, we observed a significant downregulation of superoxide dismutase (*SOD)* genes at 39 °C and 41 °C compared to the controls (*p* < 0.001), which further exposed cells to mitochondrial damage and cell death [66]. The *SOD* gene family expresses the antioxidants-superoxide dismutase proteins, key players in the entire antioxidant defense system [67]. Notably, the depletion of GSH has been also reported to decrease myogenic differentiation in murine skeletal muscle [68,69]. Moreover, this decline in GSH levels has been shown to have a negative impact on key indicators of mitochondrial function, including maximal respiration and spare respiratory capacity, which has demonstrated in several studies [18,70]. The observed results are concomitant with our former study on thermal stressed myoblasts [14].

## 4. Materials and Methods

### 4.1. Cell Culture

We used cell samples from our previous study [15]. In brief, skeletal muscle tissue from 20 piglets with normal birth weight (1.36  ±  0.15 kg) at two different ages (pool 1, *M. rhomboideus*, n = 10, day 5, female; pool 2, *M. rhomboideus*, n  =  10, day 20, female), which were produced at the pig-breeding facility of the Research Institute for Farm Animal Biology (FBN, Dummerstorf, Germany), was used. The German Landrace sows had ad libitum access to feed (Trede and von Pein, Itzehoe, Germany) and water, while the piglets were still with the sows in standard housing at the FBN Experimental Station. Piglets were killed at the FBN slaughterhouse using exsanguination after captive-bolt pistol (5 days of age) or electro stunning (20 days of age). Satellite cell isolation and the establishment and validation of two muscle cell pools (p5, n = 10; p20, n = 10) were performed, as previously described [15]. The previous isolation of satellite cells using *M. rhomboideus* tissue was described by Metzger at al [15]. These cryopreserved cells from day-5 and day-20 samples were further used for proliferation and differentiation into myotubes at different culture temperatures as shown on Figure 9a.

For the present experiments, cells from both pools stored in liquid nitrogen were defrosted and permanently cultured at 35, 37 (control), 39, and 41 °C. Cells were seeded in Geltrex™ (growth factor reduced, 1:100, Gibco Thermo Fisher, Schwerte, Germany)-coated 100 mm cell culture dishes (Sarsted, Nümbrecht, Germany) and grown in growth medium (DMEM) (Biochrom, Berlin, Germany) supplemented with 0.2-M L-glutamine (Carl Roth, Karlsruhe, Germany), 100-IU/mL penicillin (Biochrom), 100-μg/mL streptomycin (Biochrom), 2.5-μg/mL amphotericin (Sigma-Aldrich, Taufkirchen, Germany), 10% FBS (Sigma-Aldrich), and 10% donor horse serum (HS; Sigma-Aldrich)) for four days, in growth medium 2 (DMEM (Biochrom) supplemented with 0.2-M L-glutamine (Carl Roth), 100-IU/mL penicillin (Biochrom), 100-μg/mL streptomycin (Biochrom), 2.5-μg/mL amphotericin (Sigma-Aldrich), 10% FBS (Sigma-Aldrich), and 1-μM insulin (Sigma-Aldrich)) for one day and then in serum-free differentiation medium (MEM (Biochrom) supplemented with 0.2-M L-glutamine (Carl Roth), 100-IU/mL penicillin (Biochrom), 100-μg/mL streptomycin (Biochrom), 2.5-μg/mL amphotericin (Sigma-Aldrich), 1-μM insulin (Sigma-Aldrich), 1-μM cytosine β-D-arabinofuranoside (Sigma-Aldrich), 0.5-mg/mL bovine serum albumin (Sigma-Aldrich), 0.1-nM dexamethasone (Sigma-Aldrich), 0.5-μg/mL linoleic acid (Sigma- Aldrich), and 100-μg/mL transferrin (bovine holoform, Sigma-Aldrich)) for eight days. For microarray analysis, 4 × 10^5^ cells from each pool were seeded in 100 mm culture dishes (Sarsted). To measure mitochondrial and glycolytic functional changes, 3000 cells/well and 46 wells per pool per replicate (Seahorse XFp plate, OLS, Bremen, Germany) were used. To detect the ratio of reduced glutathione (GSH) to oxidized glutathione (GSSG), 1500 cells/well and 20 wells per pool per replicate were used (96-well microplates, Sarstedt). Three replicates were generated for each experiment.

### 4.2. RNA Isolation and Microarray Analysis

After eight days of differentiation, total RNA was extracted using TRI reagent (Sigma-Aldrich) and an RNeasy Mini Kit (Qiagen, Hilden, Germany) according to the manufacturer’s instructions. Microarray analysis were performed using Porcine Snowball Microarrays (Affymetrix, Thermo Fisher Scientific, Schwerte, Germany) containing 47,880 probe-sets. A total of 500 ng total RNA was used for cDNA synthesis and then biotin labelled with an Affymetrix WT plus Expression Kit (Affymetrix) and Genechip WT terminal labelling and hybridization Kit (Affymetrix) following the manufacturer’s instructions. The microarrays were hybridized with 24 tagged cRNA samples (4 temperature × 2 pool (p5 and p20) × 3 repeated experiment). Subsequently, cleaning and scanning were carried out, in accordance with the manufacturer’s guidelines. Affymetrix GCOC 1.1.1 software was used for quality control. Expression Console software was used for RMA (robust multichip average) normalization, with the DABG (detection above background) method used to find present genes. Probe sets with low signals that were present in fewer than 80% of the samples at each temperature were eliminated. After filtering, 13,226 probe sets were chosen for further analysis. Principal component analysis (PCA) was used for 13,226 probe set data. There was a stronger separation between temperatures than age, as shown in Figure 9b.

### 4.3. Analysis of Differentially Expressed Genes (DEGs)

Using JMP Genomic’s mixed model analysis, DEGs analysis was carried out (Version 9, SAS Institute Inc., Cary, NC, USA). As fixed variables, the temperature (35, 37, 39, or 41 °C), the pool (p5 or p20), and their interactions were employed. Using Tukey–Kramer tests, differences between least square means (LSM) were investigated. A 10% false discovery rate (FDR) was used as a threshold.

### 4.4. Functional Annotation of Gene Ontology (GO) and Analysis of Kyoto Encyclopedia of Genes and Genomes (KEGG) Pathway Enrichment 

For functional annotation of GO and analysis of KEGG pathway enrichment, we used the R package “clusterProfiler version 4.5.2.001” [71]. Each gene cluster’s enriched functional categories were calculated and compared using the enrichGO function in the clusterProfiler package. To see the differences in enriched functional categories between the clusters, we created a tree plot in which gene sets with high similarity tend to cluster together and a heatmap using adjusted *p*-values.

### 4.5. Validation of Microarray Results

The validation of the microarray results was performed using multiplex quantitative real-time PCR (RT-qPCR). Multiplex RT-qPCR was conducted using a Biomark HD system and EvaGreen fluorescence dyes (Bio-Rad, Hercules, CA, USA), as previously described [72,73]. In brief, 48.48 Fluidigm gene expression biochips were first primed in the MX integrated fluidic circuit (IFC) controller (Fluidigm, South San Francisco, CA, USA), before being loaded with the pre-amplified cDNA samples and eventually analyzed using a Biomark HD instrument (Fluidigm, CA, USA). The raw RT-qPCR results were retrieved with instrument-specific analysis software (v. 3.0.2; Fluidigm, CA, USA). The Ct’s geometric means of the reference genes *RN18S*, *5S_rRNA*, *RPL32,* and *pACTB* were used for data normalization. The primer list used for validation of microarray data can be found in Appendix A. The statistical analyses of qPCR data were calculated using SAS (Version 9.4, SAS Institute Inc.). Pearson’s correlation coefficient (r) analysis was performed with R Studio v.4.3.

### 4.6. Mitochondrial Bioenergetics Assay and Ratio of Reduced/Oxidized Glutathione

A GSH/GSSG-GloTM Assay Kit (Promega, Walldorf, Germany) was used in accordance with the manufacturer’s instructions for adherent cells, to measure the ratio of reduced glutathione (GSH) and oxidized glutathione (GSSG). The mitochondrial and glycolytic functions of the cells were analyzed using a Seahorse XFp Extracellular Metabolic Flux Analyzer, following the standard manufacture protocol. The mitochondrial functions were assessed based on several parameters, including oxygen consumption rate (OCR, pmol/min/µg protein), which was measured to account for non-mitochondrial respiration, basal respiration, maximal respiration, proton leak, ATP production, and spare respiratory capacity. Data were submitted for analysis of variance using the MIXED method in SAS for statistical analysis (version 9.4, SAS Institute Inc.). Three fixed parameters were used: pool (p5 or p20), temperature (35, 37, 39, or 41 °C), and the interaction of temperature and pool. Tukey–Kramer tests were used to assess differences between the LSMs, and *p* ≤ 0.05 was defined as a significant.

### 4.7. Data Visualisation

Heatmaps for visualizing the Z-score of selected genes were generated using Heatmaply v.1.4.2 for R studio v. 4.3. [74]. Pathview Web, offered as a web service, was used to create KEGG pathway visualization (https://pathview.uncc.edu/, accessed on 9 January 2023) [75]. Volcano plots were generated using EnhancedVolcano v 1.18.0 for R studio v. 4.3. [76]. Network representation of qPCR selected genes was designed using STRING (Search Tool for the Retrieval of Interacting Genes/Proteins), offered as a webtool (https://string-db.org/, accessed on 1 July 2023) [77].

## 5. Conclusions

Our investigation emphasized the transcriptional changes and metabolic responses of thermal stress in differentiating myotubes derived from piglets with different thermoregulatory capacities (p5 thermolabile, p20 thermostabile) at different ages. Both cold stress and heat stress showed common KEGG pathway involvement, which underpins the similar response against the different stimuli. Nevertheless, the impacts of CS and HS alter the mitochondrial function, which was more prominent during permanent culture of muscle cells above 37 °C. Analysis of differentiating myotubes from different developmental stage donors showed a prominent effect on DEGs, mainly linked to muscle development and differentiation, particularly for CS temperatures, probably caused by thermo-adaptability. A comparison with the previous study on myoblasts performed by our group showed specific pathway activation in myotubes, which undergo a deep transcriptome reshaping after the differentiation process. Nevertheless, the expression of specific genes such as the *RMB* and *HSP* gene families followed a similar pattern, indicating the preservation of mechanisms against thermal stress. Overall, we observed that the effect of CS related to the number of DEGs or biological pathways during differentiation of myotubes was more dominant than HS. These results can be traced back to the original cell donor (p20 vs. p5) and their capacity for thermoregulation. Interestingly, the reduction in GSH concentration provided further insights into the mechanism of myogenic differentiation during HS, which is strictly dependent on the redox metabolism and availability of specific enzymes.

The current study provides insightful data on the specific genes involved, biological processes, and regulation of metabolic pathways affecting myogenic differentiation mechanisms under temperature stress from donor cells with different thermoregulatory capacities, which can be used to develop thermal management strategies to enhance skeletal muscle growth.

## Figures and Tables

**Figure 1 ijms-24-13599-f001:**
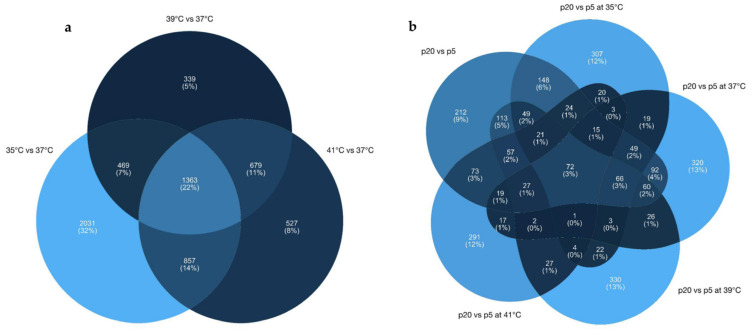
(**a**) Venn diagrams of overlapping DEGs between different experimental temperatures. (**b**) Venn diagrams of overlapping DEGs between different pools exposed at different experimental temperatures.

**Figure 2 ijms-24-13599-f002:**
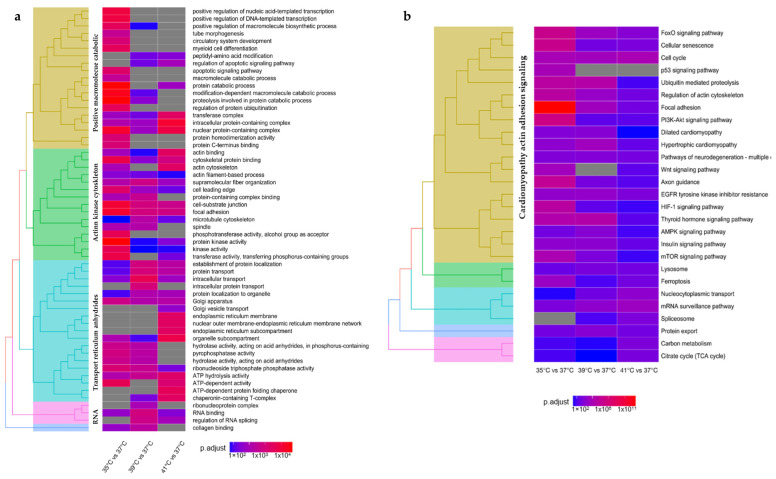
(**a**) Clustered enriched gene ontology (GO) terms of biological processes (BP), molecular function (MF), and cellular component (CC) assigned to myotubes permanently cultured at 35°, 39°, or 41 °C compared to 37 °C. The heatmap color coding indicates the adjusted *p*-value. (**b**) Clustered enriched gene ontology (GO) terms from the Enriched Kyoto Encyclopedia of Genes and Genomes (KEGG) pathways assigned to myotubes permanently cultured at 35°, 39°, or 41 °C compared to 37 °C. The heatmap color coding indicates the adjusted *p*-value.

**Figure 3 ijms-24-13599-f003:**
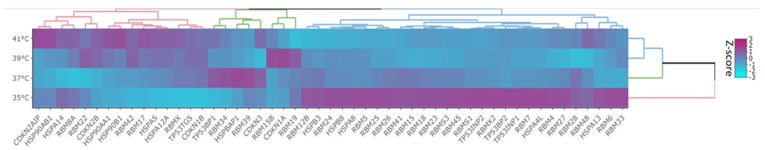
Heatmap of selected DEGs involved in cell cycle regulation, thermal stress response, and development regulation. The groups of genes with a similar expression pattern at each temperature were classified as red, green, and blue colors and these genes belong to cell cycle (p53/CDNK), heat shock proteins (*HSPs*), developmental regulators genes, signal transduction gene markers, and RNA binding protein family (*RBM*). The heatmap was generated using the hierarchical clustering method in Heatmaply version 1.4.2.

**Figure 4 ijms-24-13599-f004:**
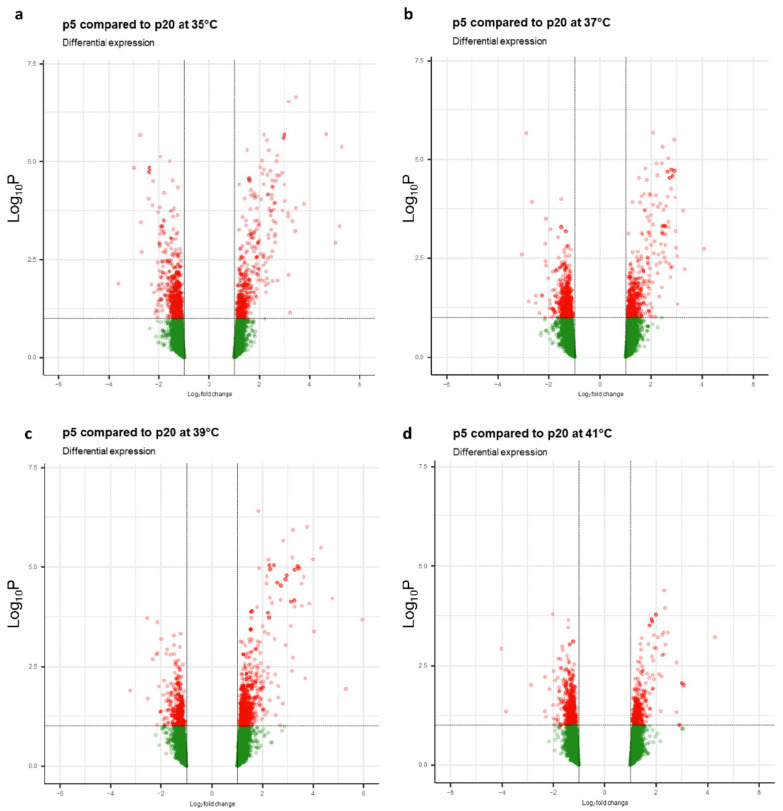
Volcano plots representing the DEGs registered between the different experimental temperatures: (**a**) p5 compared to p20 at 35 °C, (**b**) p5 compared to p20 at 37 °C, (**c**) p5 compared to p20 at 39 °C, (**d**) p5 compared to p20 at 41 °C. Black lines with horizontal (FDR ≤ 0.10) and vertical (FC: ± 1.1) marks represent the double filtering parameter.

**Figure 5 ijms-24-13599-f005:**
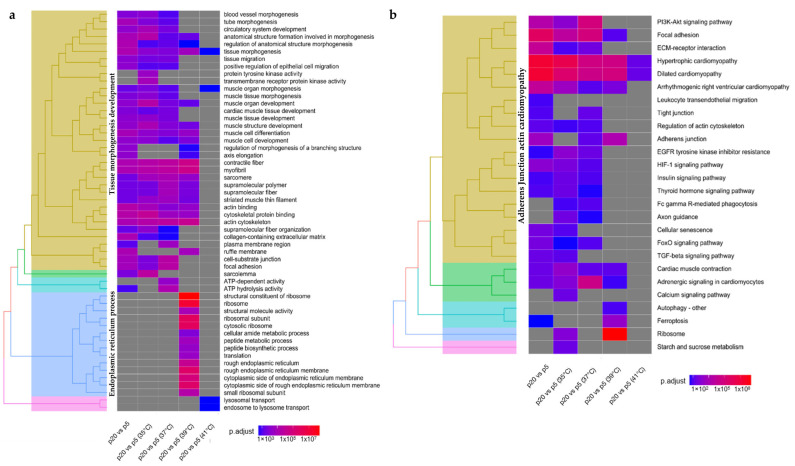
Clustered functional annotation enrichment analysis of differentially expressed genes (DEGs) between p5 and p20 at different temperatures. The analysis was conducted through (**a**) gene ontology (GO) and (**b**) Kyoto Encyclopedia of Genes and Genomes (KEGG) pathway enrichment analysis. The heatmap color coding indicates the adjusted *p*-value.

**Figure 6 ijms-24-13599-f006:**
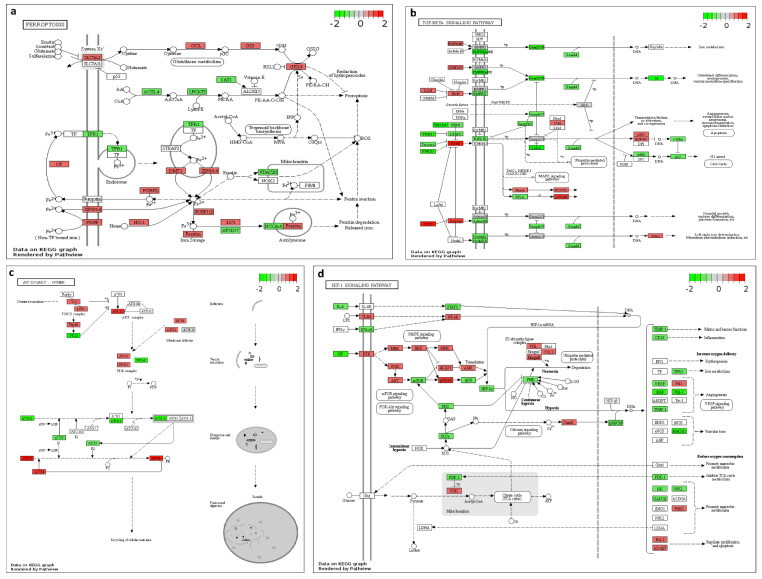
Modified KEGG pathway for (**a**) ferroptosis pathway (p5 compared to p20 at 39 °C), (**b**) TGF−beta signaling (p5 compared to p20 at 35 °C), (**c**) autophagy (p5 compared to p20 at 39 °C), (**d**) HIF−1 signaling pathway (p5 compared to p20 at 35 °C). Red denotes elevated mRNA fold change and brilliant green downregulated mRNA fold change.

**Figure 7 ijms-24-13599-f007:**
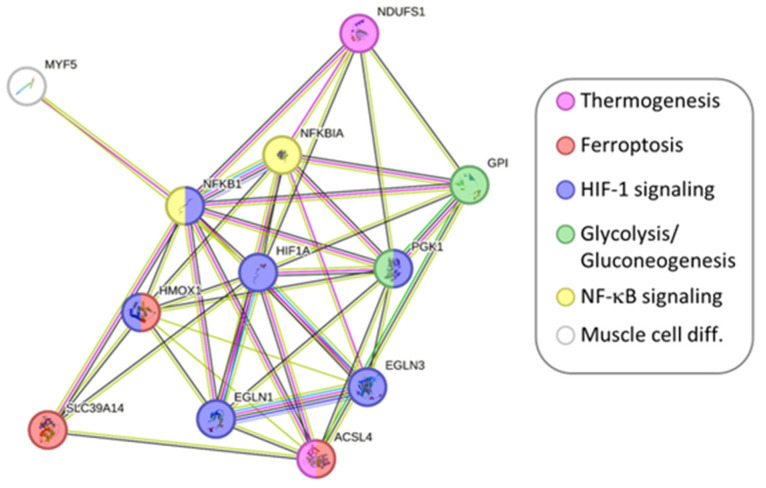
String interaction network of selected genes for microarray validation. The legend describes in which KEGG pathway they are involved.

**Figure 8 ijms-24-13599-f008:**
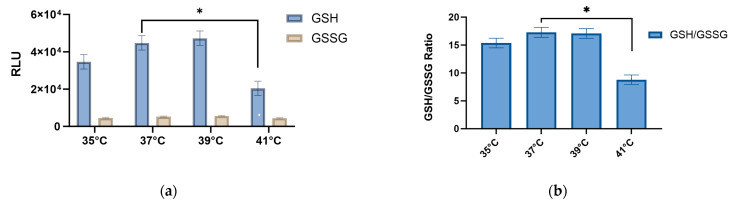
(**a**) GSH and GSSG RLU at 35°, 37°, 39°, and 41 °C. (**b**) GSH/GSSG ratio at 35°, 37°, 39°, and 41 °C. (* *p* < 0.001 when comparing between 37° and 41 °C).

**Figure 9 ijms-24-13599-f009:**
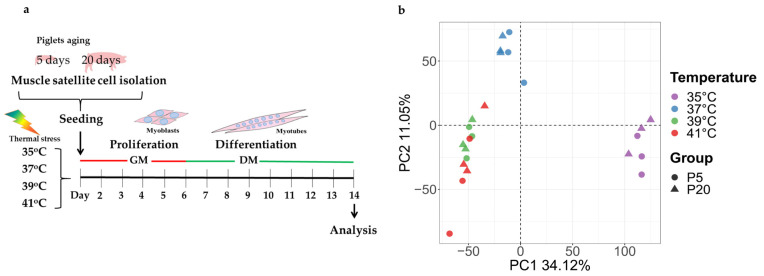
(**a**) Overview of the experimental design for the cultivation of porcine primary muscle cells. Cell pools from piglets of two different ages (5 and 20 days) were cultured at different temperatures (35, 37, 39, and 41 °C). After eight days of differentiation into myotubes, the cells were used for analyses. GM = growth medium 1 and 2, DM = differentiation medium. (**b**) PCA plot demonstrates all microarray samples used. Samples were colored based on temperature and symbols as aging (p5 and p20).

## Data Availability

The expression data are available in the Gene Expression Omnibus public repository with the GEO accession number (GSE232500: GSM7349153—GSM 7349176).

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
