# Peer review of "Transcriptomic Response of Differentiating Porcine Myotubes to Thermal Stress and Donor Piglet Age"

_ijms, 2023, doi:10.3390/ijms241713599_

Round 1

Reviewer 1 Report

This article is well designed and the data is relatively rich. However, I have the following 2 minor suggestions:

1. The quality of the pictures of the article needs to be improved;

2 I suggest to detect the expression of the relevant protein in KEGG in Fig 6 so that we can draw conclusions with more certainty.

Reviewer 2 Report

In this manuscript, the authors investigated the effects of thermal stress on differentiated myotubes derived from day 5 and day 20 old pig in vitro skeletal muscle satellite cells. 

Overall, the experiment design is confusing, as I did not see the point why the authors used the in vitro differentiated myotubes versus the in vivo differentiated myotubes, it should be very easy to isolate the multinucleated myotubes from other mononuclear cells in porcine skeletal muscle tissue. Even if the authors use the in vitro system due to the inconvenient control of the pig house’s temperature. The authors should know that it is different to treat the cell during the process of myogenesis versus to directly treat the differentiated myotubes using different temperature, if the authors wanted to study the “Transcriptomic Response of Differentiating Porcine Myotubes to Thermal Stress and Donor Piglet Age” as described in the title of this manuscript, they should treat the differentiated myotubes using different temperature, rather than treat the cell during the whole myogenesis process. The results could make quite different. By the way, if you treat the cell for ~8 days in vitro using the different temperature, they should not described as day 5 and day 20 old cells, as you should added the treatment time. Besides the authors found a batch of DEGs, GO pathways and KEGG, I did not see too much interesting findings for this study.

Other comments are needed to address as well:

Line 66-69, rewrite this sentence, it is complex for understanding.

Results 2.1, indicate in which day of pig’s satellite cells were used for this part of experiments, D5 or D20?

Indicate the different groups of genes in Figure 3.

Line 154, change “registered” to “identified”.

Line 156-157, indicate the temperature conditions in the text.

Line 168-169, it does make sense to show the overlapped DEGs, what do you want to convey to reader? Do you mean these genes are the more steady genes, which could be differentially expressed in both cold and hot stress temperature?

Line 390-392, indicate the breed of pig used for cell isolation.

Line 409 and 410, indicate how did you collect the myotubes on the top of remaining undifferentiated cells?

Author Response

'please see the attachment.'

Reviewer 3 Report

This is attached to a file

These have been pointed out in the attached file.

Author Response

'Please see the attachment'

Reviewer 4 Report

Dear Authors, in my opinion, the manuscript seems to be interesting and presents information which is significant due to climate warming. However, in 2022 you published a highly similar manuscript with differences in cultivation (3 and here 8 days), so multiple methodological sections are highly similar; therefore they should be cited, not repeated in many cases. Moreover, I couldn't find the justification for why the first manuscript was 3 and now are 8 days, which processes are activated between 3 and 8 days of cultivation, and why this period is important. In the discussion section, you should make a comparison between the results observed in your previous study and in the present, which will explain the differences in the cultivation between 3 and 8 days. Seeing from the nature side, it should be explained which experiment better reflect the processes which can occur in vivo.  

minor corrections:

1. In the introduction - could you add in one sentence in line 57 information about the role of skeletal muscle during thermoregulation of newborn piglets, I know that there are citations but it would be better to add this information to the manuscript body.

2. methods and marital section is well described.

 one ask:  why did you choose three technical replicates when in your previous work was six? what is better

3. in the results is a lack of heatmap or PCA plot for all samples across DEGs, where particular technical replicates should be clustered in the same groups.

4. in section 2.3. are mistakes in the GSH GSSG shortcut please check between 229-234

5. line 240 OCR -  the abbreviation is not described

Author Response

'Please see the attachment'

Round 2

Reviewer 2 Report

The authors have addressed my concerns. 

Author Response

Thank you for your time and suggestions.

Reviewer 3 Report

IJMS 2567882 Peer Review Report-v2

The authors devoted quality time and addressed the initial concerns raised in the first round of review of the manuscript. To further improve it, I have suggested some minor corrections and areas for their attention.

Lines 16,17 and 413 Musculus rhomboideus and M. rhomboideus should be italicised.

Line 23 The authors should italicise the genes HSP and RBM, this should be done in the whole manuscript for these and any other genes. For instance, ADGR in line 26,

Line 24 The word ‘response’ does not seem to agree with the sentence, it should be ‘responses?’ Please check. ‘That was’ is necessary here; consider removing it.

Lines 25,26 This sentence appears hard to follow and understand. It should be paraphrased. Suggestion: Moreover, thermal stress exhibits an inhibitory effect on cell cycle, with a more pronounced downregulation during cold stress driven by ADGR genes.

Line 29 Temperatures does not agree with the other words in the sentence; consider deleting letter ‘s’.

Line 32 The word ‘determines’ does not fit in here; consider deleting letter ‘s’ from it.

Line 39 The word ‘changes’ does not fit in here; consider deleting letter ‘s’ from it.

Line 47 the article ‘a’ is missing between As and counterpart; consider adding it.

Line 71 check the spelling of ‘appropiate’.

Line 112 Temperature does not agree with the other words in the sentence; consider adding letter ‘s’ to it.

Line 142 The word ‘as’ does not fit in here; consider changing that preposition to ‘at’.

Line 165 A verb ‘are’ is missing just before the word upregulated; consider adding it.

Line 403 Cells does not agree with the other words in the sentence; consider deleting letter ‘s’.

Line 411 change day (5 and 20) to a plural noun.

These have been included in the attachment.

Author Response

'Please see the attachment'

Reviewer 4 Report

Dear authors, I see that you addressed all my suggestions and explained inaccuracies. I think that the manuscript can be published in its present form.

Author Response

(The authors gave the same response as above.)
